# Quasi-Targeted Metabolomics Approach Reveal the Metabolite Differences of Three Poultry Eggs

**DOI:** 10.3390/foods12142765

**Published:** 2023-07-20

**Authors:** Yan Wu, Hongwei Xiao, Hao Zhang, Ailuan Pan, Jie Shen, Jing Sun, Zhenhua Liang, Jinsong Pi

**Affiliations:** 1Institute of Animal Husbandry and Veterinary, Hubei Academy of Agricultural Science, Wuhan 430064, China; wuyanwh@163.com (Y.W.); xiaohongwei2003@163.com (H.X.); 15172520011@163.com (H.Z.); panailuan08319@163.com (A.P.); 1711372327@163.com (J.S.); sunjing8866@hbaas.com (J.S.); liangzhenhua2046@163.com (Z.L.); 2Hubei Key Laboratory of Animal Embryo and Molecular Breeding, Wuhan 430064, China

**Keywords:** chicken egg, duck egg, quail egg, metabolomics, metabolite

## Abstract

As a food resource and nutrient, eggs play an important role in reducing malnutrition and improving the health status around the world. We studied the metabolite profile of three kinds of eggs using a widely-targeted metabolomics-based technique to better understand the difference in metabolites among chicken, duck, and quail eggs. We identified 617 metabolites, of which 303, 324, 302, 64, 81, and 80 differential metabolites were found by two group comparisons: quail egg yolk (QY) vs. quail egg albumen (QW), chicken egg yolk (HY) vs. chicken egg albumen (HW), duck egg yolk (DY) vs. duck egg albumen (DW), quail egg (Q) vs. duck egg (D)/chicken egg (H), and duck egg (D) vs. chicken egg (H), respectively. The Venn diagram showed that 147 metabolites were shared among the chicken, duck, and quail eggs. Additionally, the nucleotide and its derivates had the largest variations among the different types of eggs. This indicates that the flavor difference of the chicken eggs, duck eggs, and quail eggs may be related to their nucleotides and their derivates. The differential metabolites between egg yolk and albumen were primarily correlated with amino acid metabolism, protein metabolism, and immune performance. The discovery of these differential metabolites paves the way for further research on the nutritional potentials of various egg types.

## 1. Introduction

Eggs are considered as an essential food that contains almost all of the essential nutrients including proteins, lipids, vitamins, minerals, and growth factors for the embryo [1]. They are one of the most widely produced livestock products worldwide [2]. In addition, eggs are a major and affordable source of animal protein and lipids in many developing countries [3]. In more developed countries, consumers are increasingly concerned about the quality of eggs and the bioavailability of beneficial functional elements [4,5]. According to recent research, egg yolks provide a substantially greater variety of nutrients than egg whites. Egg yolk is rich in lipids (especially phospholipids), proteins, and vitamins (especially vitamins A, D, E, and B vitamins) as well as calcium, iron, and phosphorus. We obtain the necessary amino acids from over 1000 recognized albumen proteins and 150 identified yolk proteins [6]. Egg components exhibit various kinds of biological actions including antioxidant, antibacterial, immunomodulatory, anticancer, and antihypertensive properties [1,6]. Both genetic and environmental factors can affect the quantity and quality of eggs [7,8,9].

It is common knowledge that chicken, quail, and duck eggs are the three most widely eaten eggs worldwide. Due to their adaptability in different environments and evolution, they may have different egg protein patterns. According to the phylogenetic research using the whole mitochondrial genome [10] and egg white proteome [11], Japanese quail (Coturnix japonica) and chicken have a significantly closer kinship than duck. According to reports, compared to other avian egg yolks, duck egg yolks have unique benefits in terms of nutrition, bioactivities, and processing [12,13]. Proteomic analysis of different poultry eggs and the identification of their biological activities have been published [14,15]. Using a comparative proteomic approach, only a small number of distinct proteins have been detected in the yolks of chicken, duck, and quail eggs [15].

However, the egg of different species has different nutritional components and biological activities, and the types and quantities of the characteristic substances in the egg white and yolk of different varieties of poultry eggs are unclear. Metabonomics is the study of various metabolites in different tissues, which is used to detect novel metabolites or changes in the metabolite ratio in tissues [16]. Metabonomics analysis is applied in a variety of fields including biomedical [17,18], plant science [16], food science [19,20], and ecological/environmental science [21,22]. Therefore, we anticipate that a metabolomics method will provide insightful information on how the species affects the egg composition.

## 2. Materials and Methods

### 2.1. Egg Preparation

Fresh eggs of quail, chicken, and duck were harvested randomly from the Poultry Research Farm of the Hubei Academy of Agricultural Science within 24 h after being laid and utilized in this study. Six chicken eggs, six duck eggs, and six quail eggs were harvested. The eggs of different poultry were harvested in October 2022 and the albumen and egg yolk of all eggs were separated, respectively.

### 2.2. Metabolites Extraction

The albumen and egg yolk (100 mg per sample) were individually ground with liquid nitrogen, and then the homogenate was resuspended utilizing pre-cooled 80% methanol. The samples were incubated on ice for 5 min and centrifuged at 15,000 g, 4 °C for 20 min. Some of the supernatant was diluted to a final concentration containing 53% methanol with the help of liquid chromatography-mass spectrometer (LC-MS) grade water. Subsequently, the samples were transferred to a new Eppendorf tube, centrifuged at 15,000× *g*, 4 °C for 20 min, and subjected to analysis utilizing the liquid chromatography-tandem mass spectrometry (LC-MS/MS) system [23].

### 2.3. LC-MS/MS Analysis

LC-MS/MS analyses were completed utilizing an ExionLC™ AD system (SCIEX) coupled with a QTRAP^®^ 6500+ mass spectrometer (SCIEX) at Novogene Co., Ltd. (Beijing, China). Samples were injected onto a Xselect HSS T3 (2.1 × 150 mm, 2.5 μm) using a 20-min linear gradient at a flow rate of 0.4 mL/min for the positive/negative polarity mode. The eluents were eluent A (0.1% formic acid–water) and eluent B (0.1% formic acid–acetonitrile). The solvent gradient was set as follows: 2% B, 2 min; 2–100% B, 15.0 min; 100% B, 17.0 min; 100–2% B, 17.1 min; 2% B, 20 min. The QTRAP^®^ 6500+ mass spectrometer was operated in positive polarity mode with a curtain gas of 35 psi, collision gas of medium, ion spray voltage of 5500 V, temperature of 550 °C, ion source gas of 1:60, and an ion source gas of 2:60. The QTRAP^®^ 6500+ mass spectrometer was operated in negative polarity mode with a curtain gas of 35 psi, collision gas of medium, ion spray voltage of −4500 V, temperature of 550 °C, ion source gas of 1:60, and an ion source gas of 2:60.

### 2.4. Identification and Quantification of Metabolites

The Novogene database was adopted to identify the experimental samples using MRM (multiple reaction monitoring). The Q3 was adopted for the measurement of metabolites while the Q1, Q3, RT (retention time), DP (declustering potential), and CE (collision energy) were used for the metabolite identification. The LC-MS/MS data files were integrated and rectified using SCIEX OS version 1.4. The following parameters were established: the minimal peak height of 500; the signal-to-noise ratio of 5; and the Gaussian smooth width of 1. Based on the peak area of Q3 (sub ion), the MRM mode of the triple quadrupole were used for the quantitative analysis of the compounds.

### 2.5. Data Analysis

KEGG (Kyoto Encyclopedia of Genes and Genomes) (http://www.genome.jp/kegg/, accessed on 19 November 2021), HMDB (Human Metabolome Database) (http://www.hmdb.ca/, accessed on 19 November 2021), and Lipidmaps (http://www.lipidmaps.org/, accessed on 19 November 2021) were used to annotate these metabolites. PCA (principal component analysis) and PLS-DA (partial least-squares discriminant analysis) were conducted using metaX (a flexible and comprehensive software for processing metabolomics data) [24]. U analysis (*t*-test) was adopted to determine the statistical significance (*p*-value). The metabolites with a VIP (variable important in projection) > 1 and a *p*-value of 0.05 as well as a fold change 2 or FC (fold change) of 0.5 were deemed differential. On the basis of the Log2 (FC) and −log10 (*p*-value) of metabolites, the metabolites of interest were filtered using ggplot2 in R using volcano plots.

The data were normalized with the help of z-scores of the intensity regions of differential metabolites and shown utilizing the R package Pheatmap (R 3.4.3) for clustering heat maps. The correlation between distinct metabolites was evaluated using the R function cor (method = Pearson). Statistically significant correlation between the differential metabolites were calculated by cor.mtest in R language. If the probability value was less than 0.05, it was considered significant, and R’s corrplot tool was used to generate correlation charts. Using the KEGG database, the roles of these metabolites and metabolic pathways were investigated. Diverse metabolites were enriched in their respective metabolic pathways. When the ratio x/n > y/N was fulfilled, the metabolic pathways were deemed enriched; when the *p*-value of the metabolic pathway was less than 0.05, the metabolic pathways were declared statistically significant.

## 3. Results

### 3.1. Metabolic Profiling

A quasi-targeted metabolomics analysis was performed for the comprehensive metabolic profiling of chicken eggs, duck eggs, and quail eggs based on LC-MS/MS. The metabolic characteristics of each sample were screened. In total, 617 metabolites were identified utilizing the LC-MS/MS system (Table 1) including amino acids and derivatives (20.75%), organic acids and derivatives (15.24%), nucleotides and derivatives (10.05%), fatty acyls (7.29%), carbohydrates and derivatives (6.65%), phospholipid (4.21%), carnitine (3.89%), hormones (3.40%), organoheterocyclic compounds (3.08%), bile acids (2.59%), sugar acids and derivatives (1.94%), benzoic acid and derivatives (1.94%), eicosanoid (1.78%), vitamins (1.62%), pyridine and derivatives (1.46%), benzene and substituted derivatives (1.46%), sugar alcohols (1.30%), polyamine (1.13%), phenols and derivatives (0.97%), cholines (0.97%), indole and derivatives (0.81%), compounds involved in the TCA cycle (0.81%), cinnamic acids and derivatives (0.49%), and others (6.16%) (Figure 1, Appendix A).

In the quail eggs, 615 metabolites were screened, while 614 and 613 metabolites were in the chicken and duck eggs, respectively. Among the three types of eggs, the numbers of upregulated differential metabolites were significantly higher than the downregulated differential metabolites. In the albumens, the differential metabolites between duck egg and chicken egg were the largest, while in the yolks, the differential metabolites between duck egg and quail egg were the largest.

Overall, these results indicate that the metabolic profiles in the three groups of samples were significantly different.

### 3.2. Multivariate Statistical Analysis

PCA with an unsupervised model was used to examine the discovered metabolites and analyze the overall differences between samples (Figure 2), while the highly concentrated character showed superior repeatability for each group. Two principal components, PC1 and PC2, explained 82.04% of the total variance in the PCA score plot, and both showed strong clustering within groups and differentiation between the three types of eggs. PC1 and PC2 contributed 76.45% and 5.59% of the variance, respectively. The results validated that PC1 separated the egg albumen and egg yolk, indicating that the components of the egg yolk and egg albumen strongly influenced the metabolite profiles of the egg yolk and egg albumen. Egg albumens of duck, chicken, and quail were mainly separated by PC2, and the egg yolks of duck, chicken, and quail were slightly separated by PC2, implying that they had different metabolite profiles. Additionally, 36 samples from the three kinds of eggs were assigned into six distinct groups. It was shown that each group had a distinct metabolite profile, where even the egg albumen and egg yolk of the same kind of eggs had a distinct metabolite profile.

The PCA results revealed that the egg albumen and egg yolk of the three types of eggs were separated into six unique groups, implying that the metabolite profiles of each group were significantly different from one another.

### 3.3. Identification of Differential Metabolites

The PLS-DA model was adopted to establish a relationship model between the relative quantitative value of the metabolites and the sample category to achieve the prediction of the sample category. Metabolites responsible for the changes were identified by pairwise comparisons of the egg albumen and egg yolk from the three different types of eggs using the PLS-DA model.

The PLS-DA scatter scores of the groups are listed in Figure 3a–l. The model evaluation parameters (R2, Q2) were harvested through 7-fold cross-validation (seven cycle cross-validation). If R2 and Q2 are closer to 1, it indicates that the model is more stable and reliable (Appendix A). These results show that the PLS-DA model was sufficiently predictable and was not over-fitted. The egg albumen and egg yolk of the three kinds of egg and the chicken eggs, duck eggs, and quail eggs were all distinctly separated from each other, indicating significantly different metabolic phenotypes and metabolic profiles among chicken eggs, duck eggs, and quail eggs.

Additionally, variable importance in projection (VIP) > 1, fold change (FC) ≥ 1.5 or FC ≤ 0.5, and differential metabolites were chosen between different groups. The screening results are displayed in Table 1 and Figure 4a–l. There were 108 significantly differential metabolites between DW and HW (57 upregulated and 51 downregulated), 100 between DW and QW (46 upregulated and 54 downregulated), 92 between HW and QW (35 upregulated and 57 downregulated), 94 between DY and HY (63 upregulated and 31 downregulated), 112 between DY and QY (56 upregulated and 56 downregulated), 99 between HY and QY (32 upregulated and 67 downregulated), 300 between DY and DW (258 upregulated and 42 downregulated), 322 between HY and HW (276 upregulated and 46 downregulated), 302 between QY and QW (262 upregulated and 40 downregulated), 48 between H and D (27 upregulated and 21 downregulated), 68 between Q and D (54 upregulated and 14 downregulated), and 67 between Q and H (58 upregulated and nine downregulated) (Appendix A). The differential metabolites for the twelve comparison groups (DW vs. HW, DW vs. QW, HW vs. QW, DY vs. HY, DY vs. QY, HY vs. QY, DY vs. DW, HY vs. HW, QY vs. QW, H vs. D, Q vs. D, and Q vs. H) were classified into 23, 23, 20, 22, 24, 22, 35, 34, 33, 18, 21, and 18 different categories, respectively (Appendix A). These results indicate that the metabolites that caused the differences among eggs were quite different. For HY vs. HW, QY vs. QW, and DY vs, DW, the metabolites with the most significant differences for the upregulated were L-Lysine, L-Lysine, and LysoPC 14:0; the metabolites with the most significant differences for the downregulated were UDP-N-acetyl-alpha-D-mannosamine, uric acid, and UMP. These results indicate that the egg yolk had a better flavor than the albumen, which was perhaps caused by the amino acid.

The metabolites of chicken eggs (H) and duck eggs (D) were compared, and 48 differential metabolites were identified. There were 27 metabolites that were upregulated including fatty acyls, nucleotide and its derivates, hormones, carbohydrates and their derivatives, sugar alcohols, organic acid and its derivative, phospholipid, organoheterocyclic compounds, phenols and their derivative, bile acids, and vitamins (the metabolite with the most significant differences was glycochenodeoxycholic acid), and 21 metabolites were downregulated including carnitine, organic acid and its derivative, nucleotide and its derivates, carbohydrates and their derivatives, sugar alcohols, sugar acids and their derivatives, polyamine, indole and its derivates, hormones, and fatty acyls (the metabolite with the most significant differences was decanoylcarnitine) (Appendix A). The metabolites of the quail eggs (Q) and duck eggs (D) were compared, and 68 differential metabolites were harvested. There were 54 metabolites that were upregulated including nucleotide and its derivates, fatty acyls, organic acid and its derivatives, benzoic acid and its derivatives, carbohydrates and their derivatives, sugar acids and their derivatives, bile acids, eicosanoid, organoheterocyclic compounds, pyridine and its derivatives, compounds involved in the TCA cycle, amino acid and its derivatives, hormones, phenols and its derivatives, phospholipid, sugar alcohols, and vitamins (the metabolite with the most significant difference was D-galactonic acid), and 14 metabolites were downregulated including carbohydrates and their derivatives, nucleotide and its derivates, amino acid and its derivatives, carnitine, eicosanoid, fatty acyls, hormones, indole and its derivatives, organic acid and its derivatives, polyamine, and sugar acids and their derivatives (the metabolite with the most significant differences was citramalate) (Appendix A). The metabolites of quail eggs (Q) and chicken eggs (H) were compared, and 67 differential metabolites were harvested. There were 58 metabolites that were upregulated (including nucleotide and its derivates, organic acid and its derivatives, amino acid and its derivatives, carnitine, fatty acyls, pyridine and its derivatives, benzoic acid and its derivatives, carbohydrates and their derivatives, sugar acids and their derivatives, eicosanoid, organoheterocyclic compounds, alcohols and polyols, organic nitrogen compounds, steroids and steroid derivatives, sugar alcohols and vitamins (the metabolite with the most significant differences was carnitine-C14), and nine metabolites were downregulated including fatty acyls, hormones, nucleotide and its derivates, organic acid and its derivatives, phospholipid and sugar alcohols (the metabolite with the most significant differences was riboflavin-5-phosphate) (Appendix A). The above-mentioned results show that nucleotide and its derivates had the largest number of variations among the different types of eggs.

There may be a variety of factors that contribute to the dissimilar metabolites in chicken eggs and duck eggs including nucleotides and their derivatives (20.83%), carnitine (12.5%), carbohydrates and their derivatives (10.42%), fatty acyls and their derivatives (10.42%), and organic acids and their derivatives (10.42%). Nucleotide and its derivates (29.41%), organic acid and its derivatives (8.82%), fatty acyls (8.82%), and carbohydrates and their derivatives (7.35%) could account for the different metabolites identified in quail eggs and duck eggs. Nucleotide and its derivates (23.88%), organic acid and its derivatives (11.94%), fatty acyls (8.96%), amino acid and its derivatives (7.46%), carnitine (7.46%), and pyridine and its derivatives (7.46%) could account for the different metabolites discovered in quail eggs and chicken eggs.

A Venn diagram was constructed to describe the relationship between joint differently metabolites among the chicken, duck, and quail eggs (Figure 4m). In the three types of eggs, the Venn diagram showed that 147 overlapping differential metabolites were shared among the comparison groups including amino acid and its derivatives (52), organic acid and its derivatives (12), phospholipid (9), carbohydrates and their derivatives (8), fatty acyls (6), nucleotide and its derivates (6), bile acids (5), benzene and substituted derivatives (4), carnitine (4), organoheterocyclic compounds (4), vitamins (4), and others (21). A total of 59, 56, and 47 metabolites were tested only in the quail, chicken, and duck eggs, respectively.

### 3.4. KEGG Annotation and Enrichment Analysis of Differential Metabolites

The KEGG database is a widely-selected tool to analyze the signal transduction pathways and metabolite accumulation [25]. Here, we annotated and enriched the different metabolites from each comparison group, and then we separated them into their respective KEGG pathways.

The differential metabolites in the DW/HW vs. QW, DW vs. HW, DY/HY vs. QY, DY vs. HY, DY vs. DW, HY vs. HW, QY vs. QW, H/Q vs. D and Q vs. H groups enriched to the KEGG database were implicated in 50, 51, 54, 57, 56, 52, 90, 78, 62, 48, 39, and 46 pathways, respectively, and the major pathways are described in bubble plots (Figure 5a–l, Appendix A). The metabolic pathways associated with the pentose phosphate pathway, galactose metabolism, prostate cancer, and pathways in cancer were significantly enriched (*p* < 0.05) when comparing H vs. D. When comparing Q vs. D, the significant enrichment metabolic pathways were only related to prostate cancer (*p* < 0.05); when comparing Q vs. H, purine metabolism and the glucagon signaling pathway were the significantly enriched metabolic pathways (*p* < 0.05); when comparing DW vs. HW, inositol phosphate metabolism, galactose metabolism, citrate cycle (TCA cycle), metabolic pathways, and glycerolipid metabolism were the significantly enriched metabolic pathways (*p* < 0.05); when comparing DW vs. QW, pyruvate metabolism and insulin resistance were the significantly enriched metabolic pathways (*p* < 0.05); when comparing HW vs. QW, purine metabolism, pyruvate metabolism, and the pentose phosphate pathway were the significantly enriched metabolic pathways (*p* < 0.05); when comparing DY vs. HY, prostate cancer was the only significantly enriched metabolic pathway (*p* < 0.05); when comparing DY vs. QY, purine metabolism, C5-branched dibasic acid metabolism, and olfactory transduction were significantly enriched metabolic pathways (*p* < 0.05); when comparing HY vs. QY, purine metabolism, the cGMP-PKG signaling pathway, olfactory transduction, alcoholism, and antifolate resistance were the significantly enriched metabolic pathways (*p* < 0.05); when comparing DY vs. DW, mineral absorption, protein digestion and absorption, aminoacyl-tRNA biosynthesis, central carbon metabolism in cancer, 2-oxocarboxylic acid metabolism, valine, leucine and isoleucine biosynthesis, porphyrin and chlorophyll metabolism, and the biosynthesis of amino acids were the significantly enriched metabolic pathways (*p* < 0.05); when comparing HY vs. HW, aminoacyl-tRNA biosynthesis, protein digestion and absorption, biosynthesis of amino acids, and mineral absorption were the significantly enriched metabolic pathways (*p* < 0.05); when comparing QY vs. QW, aminoacyl-tRNA biosynthesis, glycine, serine and threonine metabolism, protein digestion and absorption, and lysine degradation were the significantly enriched metabolic pathways (*p* < 0.05).

Collectively, there could be different metabolite profiles in the three types of eggs and the different egg albumens or egg yolks.

## 4. Discussion

### 4.1. Metabolites Identified in the Three Types of Eggs

Eggs are an indispensable source of nourishment for human survival. Eggs provide a nearly perfect mix of important elements including n-3 fatty acids (FAs), vitamin E, selenium, and so on [26]. These nutrients provide protection against several chronic conditions [27]. According to previous research, the egg yolk comprises around 48% water, 34% lipids, and 17% proteins, whereas the egg albumen has approximately 88% water, 11% proteins, and 0.02% lipids [28,29]. On the basis of liquid and gas chromatography, the concentration of tiny molecular nutrients in eggs including amino acids and fatty acids has also been documented [30,31].

Chicken eggs are a very nutritious food item that has been eaten by humans as a rich source of nutrients since ancient times [32,33]. Eggs are an excellent source of amino acids and vitamins B2, B12, and D. In addition to minerals like calcium and iron, chicken eggs include vital nutrients such as choline and phosphoryl lipids [34]. In the chicken egg yolk, amino acid concentrations are higher than those of other metabolites; alternatively, chicken albumen has a greater quantity of sugar than any other metabolite [35]. Egg whites include 10% high-molecular-weight substances such as proteins and glycoproteins, and 0.7% free sugars such as glucose and fructose [33]. The second most popular egg in the world are duck eggs [36]. The primary components of duck eggs are lipids and proteins, making them an excellent source of daily nutrition [37]. Duck eggs are a great food choice since they include a wide variety of nutrients including fat-soluble vitamins, polyunsaturated fatty acids, proteins, minerals, and more. Amino acids, carbohydrates, and lipids are the primary metabolites in duck egg yolks, whereas amino acids, benzene, and indoles make up the albumen [38]. Quail eggs are a kind of food that is inexpensive, of high quality, and nutritionally dense. They are often consumed by humans and are referred to as “the ginseng of animals” [39]. Regular eating of quail eggs has been demonstrated to combat a number of ailments, boost immunity, support memory health, increase brain activity, and balance the nervous system. Tokuşolu et al. found that quail eggs contained more antioxidants, minerals, and vitamins than other types of eggs [40]. Similar physiological activities to those of regular egg whites are performed by the albumen of quail eggs including “buffering and damping, safeguarding embryos, and giving nutrients for embryonic development.” Additionally, prior research has shown that the protein level of freeze-dried quail egg albumen is greater than that of chicken egg albumen protein [41]. However, there has been no published study on the differences in metabolites between the three egg varieties.

In this work, we analyzed the metabolomes of chicken egg, duck egg, and quail egg to provide a complete metabolic profile for each. Among the two pairwise comparisons in Q vs. D/H, 147 overlapping differential metabolites were confirmed as the key metabolites of quail eggs including amino acid and its derivatives (52), organic acid and its derivatives (12), phospholipid (9), carbohydrates and its derivatives (8), fatty acyls (6), nucleotide and its derivates (6), bile acids (5), benzene and substituted derivatives (4), carnitine (4), organoheterocyclic compounds (4), vitamins (4), and others (21). A total of 59, 56, and 47 metabolites were tested only in the quail, chicken, and duck eggs, respectively. Based on our findings, the three eggs primarily contained amino acid and its derivatives, organic acid and its derivatives, phospholipid, carbohydrates and their derivatives, fatty acyls, nucleotide and its derivatives, bile acids, benzene and substituted derivatives, carnitine, organoheterocyclic compounds, and vitamins. The nucleotide and its derivatives exhibited the greatest number of differences among the egg types.

Nucleotides and related substances are essential biomolecules in almost all biological activities. In addition, extensive data suggest that nucleotides are crucial for immunological activity and physiological function [42]. Nucleotides are regarded as significant flavoring agents in meat products [43]. Nucleotides, which contribute significantly to the taste of pig, are utilized to distinguish various types and meat cuts [44]. Inosine-5′-monophosphate (IMP) is one of the primary taste components that impact the umami of farmed puffer fish among the nucleotides [45]. Nucleotides are often addressed in relation to animal muscle, although egg products are seldom mentioned. According to the metabolomics study in this paper, it was found that adenosine 5′-monophosphate (AMP), inosine, 1-methylguanine, and other nucleotides and their associated compounds were distinct metabolites in the three kinds of eggs. Based on the effects of nucleotides on flavor formation, nucleotides may play key roles in the umami of chicken eggs, duck eggs, and quail eggs. The nucleotides and their derivates in the categories of Q vs. H, Q vs. D, and H vs. D were 24%, 29%, and 21%, respectively; the amino acids and their derivates in the categories of Q vs. H, Q vs. D, and H vs. D were 7%, 3%, and 2%, respectively (listed in Appendix A). These results indicate that the flavor of quail eggs is better than that of chicken eggs and duck eggs, which may be related to the higher nucleotides and amino acid contents. For a more delicious taste, people can choose to eat quail eggs.

### 4.2. Metabolites Identified in the Egg Yolks and Albumens

In this research, a total of 179 differential metabolites were identified in the three comparison groups including 27, 54, and 58 upregulated and 21, 14, and 9 downregulated metabolites in the H vs. D group, Q vs. D group, and Q vs. H group, respectively (Figure 4m). Only 29 metabolites were identified from the three comparison groups, showing notable changes in the metabolite profiles of the three poultry varieties.

Metabolic pathway analysis of 147 differential metabolites validated that common differential metabolic pathways included prostate cancer, purine metabolism, and the glucagon signaling pathway, indicating a significant difference in the relative content of purine, glucagon, progesterone, and testosterone in the Q vs. D/H comparison group.

The differential metabolites in the DY vs. DW group were also enriched in mineral absorption, protein digestion and absorption, aminoacyl-tRNA biosynthesis, central carbon metabolism in cancer, 2-oxocarboxylic acid metabolism, valine, leucine, and isoleucine biosynthesis, porphyrin and chlorophyll metabolism, and the biosynthesis of amino acids (*p*-value < 0.05). The differential metabolites in the HY vs. HW group were enriched in the aminoacyl-tRNA biosynthesis, protein digestion and absorption, biosynthesis of amino acids, and mineral absorption (*p*-value < 0.05). The differential metabolites in the QY vs. QW group were enriched in aminoacyl-tRNA biosynthesis, glycine, serine, and threonine metabolism, protein digestion and absorption, and lysine degradation (*p*-value < 0.05). These results suggest that aminoacyl-tRNA biosynthesis, protein digestion and absorption, and the biosynthesis of amino acid pathways are significantly different between the egg yolk and albumen. The aminoacyl-tRNA biosynthesis pathway is crucial for reacting to danger signals and regulating immunity against viral infections [46]. The protein digestion and absorption pathway regulate the hydrolysis and transport of proteins, peptides, and amino acids [47]. Based on these, it is speculated that the differential metabolites between the egg yolk and albumen are mainly related to amino acid metabolism, protein metabolism, and immune performance.

Therefore, identifying these differential metabolites lays the foundation for future research on the function and nutritional value of various egg types.

## 5. Conclusions

In this work, a metabolomics approach based on LC-MS/MS was employed to detect various metabolites in three egg types. This study sheds light on the composition of metabolites for several egg types as they were significantly different including the albumens and egg yolks; the nucleotide and its derivatives had the greatest variation among the different egg types. The differential metabolites between the egg yolk and albumen were mostly associated with amino acid metabolism, protein metabolism, and immunological function. The identification of these distinct metabolites lays the ground-work for the future study of the function and nutritional value of various egg types.

## Figures and Tables

**Figure 1 foods-12-02765-f001:**
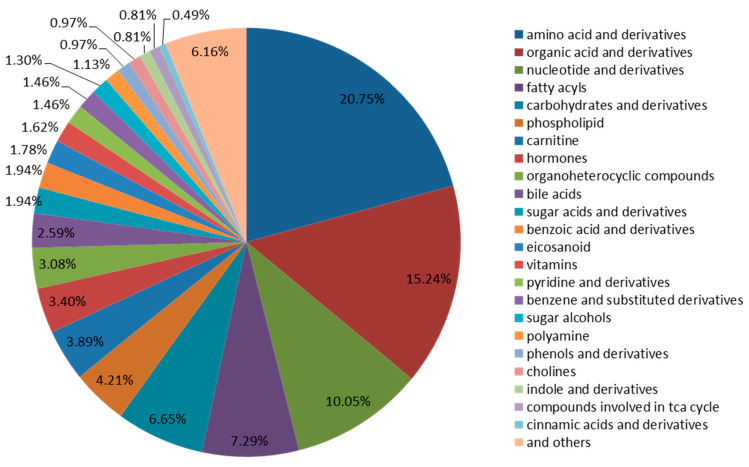
Classification of the 617 metabolites in the three types of eggs.

**Figure 2 foods-12-02765-f002:**
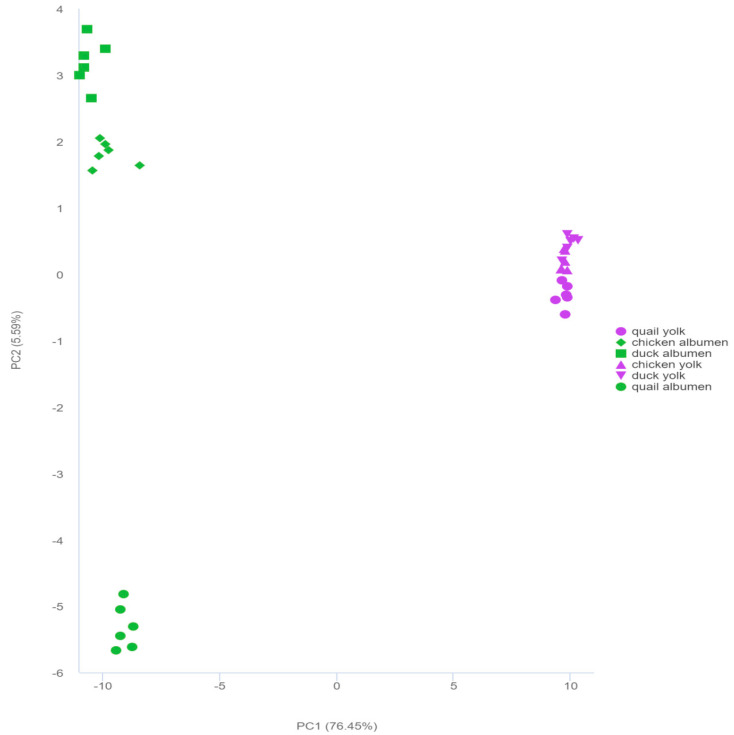
Principal component analysis (PCA) of the metabolites in the three types of egg albumen and egg yolk.

**Figure 3 foods-12-02765-f003:**
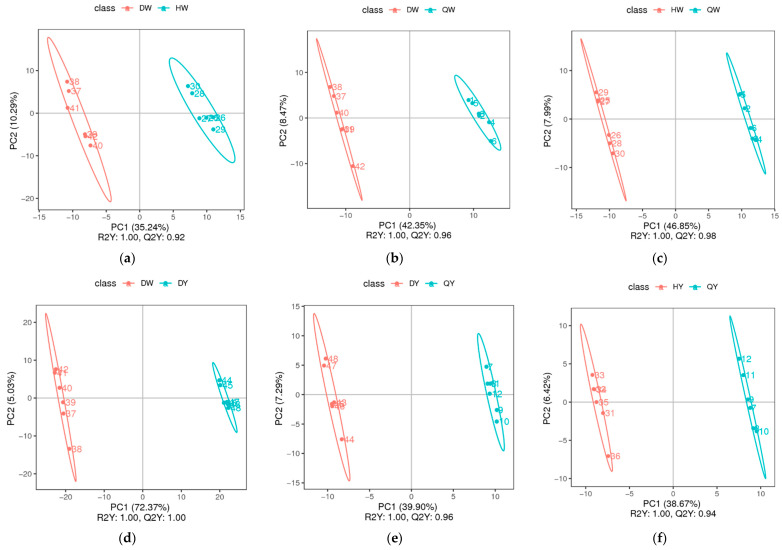
The PLS-DA scatter scores of the albumen and yolk of three types of egg. (**a**–**l**) Score scatters from PLS-DA in the DW vs. HW groups (**a**), the DW vs. QW groups (**b**), the HW vs. QW groups (**c**), the DY vs. HY group (**d**), the DY vs. QY group (**e**), the HY vs. QY group (**f**), the DW vs. DY group (**g**), the HW vs. HY group (**h**), the QW vs. QY group (**i**), the D vs. H group (**j**), the D vs. Q group (**k**), and the H vs. Q group (**l**).

**Figure 4 foods-12-02765-f004:**
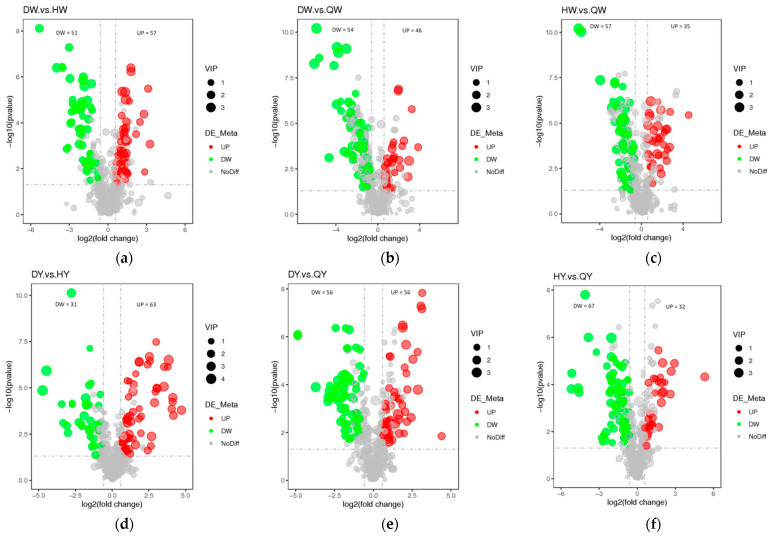
Differential metabolites analysis and the Venn diagram among the albumen and yolk of the three types of egg. (**a**–**l**) Volcano plots showing the differential metabolite expression levels between the DW vs. HW group (**a**), the DW vs. QW group (**b**), the HW vs. QW groups (**c**), the DY vs. HY group (**d**), the DY vs. QY group (**e**), the HY vs. QY group (**f**), the DY vs. DW group (**g**), the HY vs. HW group (**h**), the QY vs. QW group (**i**), the H vs. D group (**j**), the Q vs. D group (**k**), and the Q vs. D group (**l**). (**m**) Venn diagram illustrating the overlapping and specific differential metabolites for the three comparison groups (chicken egg, duck egg, and quail egg).

**Figure 5 foods-12-02765-f005:**
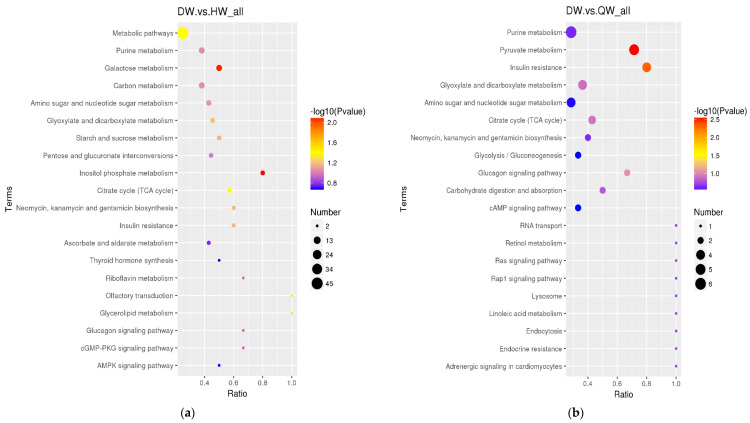
The KEGG pathway analysis of differential metabolites for different comparison groups. (**a**–**l**) KEGG pathway for the DW vs. HW group (**a**), the DW vs. QW group (**b**), the HW vs. QW group (**c**), the DY vs. HY group (**d**), the DY vs. QY group (**e**), the HY vs. QY group (**f**), the DY vs. DW group (**g**), the HY vs. HW group (**h**), the QY vs. QW group (**i**), the H vs. D group (**j**), the Q vs. D group (**k**), and the Q vs. H group (**l**).

**Table 1 foods-12-02765-t001:** Results of the metabolite difference analysis.

Compared Samples	Num. of Total Ident.	Num. of Total Sig.	Num. of Sig. Up	Num. of Sig. Down
QY. vs. QW_all	617	302	262	40
HY. vs. HW_all	617	322	276	46
DY. vs. DW_all	617	300	258	42
HW. vs. QW_all	617	92	35	57
DW. vs. QW_all	617	100	46	54
DW. vs. HW_all	617	108	57	51
HY. vs. QY_all	617	99	32	67
DY. vs. QY_all	617	112	56	56
DY. vs. HY_all	617	94	63	31
H. vs. D._all	617	48	27	21
Q. vs. D._all	617	68	54	14
Q. vs. H._all	617	67	58	9

Note: D: duck egg; DW: duck albumen; DY: duck egg yolk; H: chicken egg; HW: chicken albumen; HY: chicken egg yolk; Q: quail egg; QW: quail albumen; QY: quail egg yolk.

## Data Availability

The data presented in this study are available on request from the corresponding author. The data are not publicly available due to The metabolite database used in this analysis is owned by the Novogene Co., Ltd. and falls under the scope of core technology confidentiality.

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
