# Peer review of "Quasi-Targeted Metabolomics Approach Reveal the Metabolite Differences of Three Poultry Eggs"

_foods, 2023, doi:10.3390/foods12142765_

Round 1

Reviewer 1 Report

Overall its a nice presentation. It may find mass attention in the concerned field. There are some issues. Please address those.

1. In abstract, abbreviations should not be used. You can use an abbreviation once the full form is used

2. Abstract section can be improved

3. In 2.3 you have mentioned that HPLC MS-MS technique, in description it is LC MS-MS and in conclusion it is UPLC MS-MS. Please check the issue.

4. Add a glossary section.

5. Add future perspectives and/or your recommendations at the end of discussion section.

Minor editing should be done in the existing manuscript. 

Author Response

  1. In abstract, abbreviations should not be used. You can use an abbreviation once the full form is used

       √  We have revised the abbreviations to full form in the abstract.

  1. Abstract section can be improved.

       √  We have added some explanations about the research results in the abstract.

  1. In 2.3 you have mentioned that HPLC MS-MS technique, in description it is LC MS-MS and in conclusion it is UPLC MS-MS. Please check the issue.

        √  We have checked and revised them to LC MS-MS.

  1. Add a glossary section.

        √  We have added the glossary section after the conclusion .

  1. Add future perspectives and/or your recommendations at the end of discussion section.

       √  We had already described future perspectives and suggestions at the end of the discussion ( “As such, identifying these differential metabolites lays the ground-work for future study of the function and nutritional value of various eggs types.”). For a clearer expression, we have revised this sentence to “Therefore, identifying these differential metabolites lays the foundation for future research on the function and nutritional value of various eggs types.

Reviewer 2 Report

In their manuscript entitled "Quasi-Targeted Metabolomics Approach Revealing the Metabolite Differences of Three Poultry Eggs," Jinsong Pi and colleagues conducted a metabolomics analysis of three kinds of eggs, including chicken, duck, and quail eggs (N=6 for each type). The authors further analyzed the shared and differential metabolites among the three types of eggs and between egg yolk and albumen. They found that nucleotides and their derivatives exhibited the largest variations among different types of eggs. The differential metabolites between egg yolk and albumen were primarily correlated with amino acid metabolism, protein metabolism, and immune performance. Considering that these three types of eggs are widely consumed worldwide, it is interesting to explore the differences and shared metabolite compositions, which will enhance our understanding of the function and nutritional value of various egg types. However, there are some commands regarding the manuscript.

Major comments:

1.       Did the authors use quality control (QC) samples to correct deviations and errors caused by the instrument, as well as normalize the relative content of metabolites among samples?

2.       Could the authors explain why they used raw eggs instead of boiled eggs for the analysis? Since boiled eggs are the common form of consumption, it would be beneficial to understand if the boiling step significantly alters the metabolites.

3.       There are other associated publications on metabolite analysis. Could the authors highlight the advantages of their paper compared to existing studies?

4.       Instead of merely describing the differential metabolic pathways, the authors should provide a more in-depth discussion. For example, based on the manuscript, the authors could offer guidance on choosing egg types for different individuals, such as young people, older individuals, or those who have undergone surgery. Are there any potentially harmful metabolites that we should be aware of? Additionally, did the authors identify novel metabolites in eggs that are not widely known?

5.       In the data analysis section, all the R commands should be provided and made available online.

6.       In Figure 1, it would be more informative if the authors could provide separate classification for chicken, duck, and quail eggs, allowing for a better comparison among the three types.

7.       In Figure 2, to improve readability, the author can use the same shape but different colors or the same color but different shapes to represent yolk and albumen for each species type.

8.       In the Results section, it would be more convenient for readers if the DE (differentially expressed) numbers were placed on each volcano plot instead of describing every DE number in the main text.

9.       It is suggested to include the DE metabolites in the main text, as readers may be more interested in knowing which metabolites are differentially expressed.

10.   In the abstract, please avoid using abbreviations without providing explanations, such as QY, QW, HY, as it may hinder comprehension.

Minor comments:

1.       Please ensure consistency in writing format. For instance, the authors used both "P-value" and "p-value" in the text. Additionally, when introducing an abbreviation, please provide its full name the first time it is used, as in "as well as a fold change 2 or FC 0.5 were deemed differential."

2.       In Table 1, the column titles are too close together, resulting in some titles being covered. Furthermore, please provide a more detailed explanation in the table legend regarding the meanings of QY, QW_all, and other abbreviations.

3.       The sentence "Metabolites responsible for the changes were identified by pair-wise comparisons of egg albumen, egg yolk, egg albumen, and egg yolk from three different types of eggs using the PLS-DA model" contains duplicate words. Please carefully review the entire manuscript to avoid such mistakes.

4.       It would be more logically reasonable to place the following paragraph in the 4.1 section instead of the 4.2 section: "Quail eggs are often referred to as the 'ginseng of animals' due to their high quality and rich nutrition [39]. Among the two pairwise comparisons in Q vs. D/H, 147 overlapping differential metabolites were identified as key metabolites of quail eggs. These metabolites include amino acids and their derivatives (52), organic acids and their derivatives (12), phospholipids (9), carbohydrates and their derivatives (8), fatty acyls (6), nucleotides and their derivatives (6), bile acids (5), benzene and substituted derivatives (4), carnitine (4), organoheterocyclic compounds (4), vitamins (4), and others (21). Additionally, 59, 56, and 47 metabolites were exclusively found in quail, chicken, and duck eggs, respectively."

The English language in the manuscript is generally clear and understandable, but there is room for minor improvements to enhance clarity and readability.

Some sentences could benefit from minor revisions to improve clarity and readability.

The use of abbreviations and acronyms could be explained better to aid comprehension.

Some suggestions have been made to revise sentence structures and improve word choice for better flow and coherence.

Author Response

Major comments:

  1. Did the authors use quality control (QC) samples to correct deviations and errors caused by the instrument, as well as normalize the relative content of metabolites among samples?

        RE:  Yes, the QC samples were used. And the QC samples were made by  mixing experimental samples of equal volume.

  1. Could the authors explain why they used raw eggs instead of boiled eggs for the analysis? Since boiled eggs are the common form of consumption, it would be beneficial to understand if the boiling step significantly alters the metabolites.

        RE:  The main purpose of this study is to investigate the differential metabolites of three types of raw eggs. The boiled eggs study will be researched in the next step.

  1. There are other associated publications on metabolite analysis. Could the authors highlight the advantages of their paper compared to existing studies?

        RE:  This study mainly analyzes the differential metabolites in egg albumen and egg yolk for different type eggs, and explains the differences among the three types of eggs from the perspective of metabolites. The existing studies were mainly research the differential protein in different egg yolk and egg albumen, or differential metabolites in chicken egg yolk  or egg albumen which feeding different crops. In summary, the content and purpose are different between this research and existing studies.

  1. Instead of merely describing the differential metabolic pathways, the authors should provide a more in-depth discussion. For example, based on the manuscript, the authors could offer guidance on choosing egg types for different individuals, such as young people, older individuals, or those who have undergone surgery. Are there any potentially harmful metabolites that we should be aware of? Additionally, did the authors identify novel metabolites in eggs that are not widely known?

        RE:  we have added some guidance on choosing egg types in the discussions. Based on this paper, there are no potentially harmful metabolites. There is no novel metabolites were identified in this paper.

  1. In the data analysis section, all the R commands should be provided and made available online.

      RE:  There are too many script codes involved in the data analysis. The R language was mainly used to analyze the results. We have added the version for R.

  1. In Figure 1, it would be more informative if the authors could provide separate classification for chicken, duck, and quail eggs, allowing for a better comparison among the three types.

        RE:  The separate classification for chicken, duck, and quail eggs were listed in supplement Figure S2.

  1. In Figure 2, to improve readability, the author can use the same shape but different colors or the same color but different shapes to represent yolk and albumen for each species type.

      RE:  We have redrawn the Figure 2 with the same color but different shapes to represent yolk and albumen for each species type.

  1. In the Results section, it would be more convenient for readers if the DE (differentially expressed) numbers were placed on each volcano plot instead of describing every DE number in the main text.

      RE:  We have added the DE numbers each volcano plot and instead them in the paper.

  1. It is suggested to include the DE metabolites in the main text, as readers may be more interested in knowing which metabolites are differentially expressed.

      RE:  There were 618 DE metabolites in this paper. The table for DE metabolites is so big for listing it in the main text. So, we listed the DE metabolites as a supplemental table (Supplemental Table S1).

  1. In the abstract, please avoid using abbreviations without providing explanations, such as QY, QW, HY, as it may hinder comprehension.

      RE:  We have added the glossary after the conclusion for explaining the abbreviation which appeared in the peper.

Minor comments:

  1. Please ensure consistency in writing format. For instance, the authors used both "P-value" and "p-value" in the text. Additionally, when introducing an abbreviation, please provide its full name the first time it is used, as in "as well as a fold change 2 or FC 0.5 were deemed differential."

       RE:  We have revised the “P-value” to “p-value” in the paper and added the full name for all abbreviation.

  1. In Table 1, the column titles are too close together, resulting in some titles being covered. Furthermore, please provide a more detailed explanation in the table legend regarding the meanings of QY, QW_all, and other abbreviations.

        RE:  We have added the full name for all abbreviation and adjusted the column titles.

  1. The sentence "Metabolites responsible for the changes were identified by pair-wise comparisons of egg albumen, egg yolk, egg albumen, and egg yolk from three different types of eggs using the PLS-DA model" contains duplicate words. Please carefully review the entire manuscript to avoid such mistakes.

      RE:  We have corrected this sentence and review the entire manuscript to avoid such mistakes.

  1. It would be more logically reasonable to place the following paragraph in the 4.1 section instead of the 4.2 section: "Quail eggs are often referred to as the 'ginseng of animals' due to their high quality and rich nutrition [39]. Among the two pairwise comparisons in Q vs. D/H, 147 overlapping differential metabolites were identified as key metabolites of quail eggs. These metabolites include amino acids and their derivatives (52), organic acids and their derivatives (12), phospholipids (9), carbohydrates and their derivatives (8), fatty acyls (6), nucleotides and their derivatives (6), bile acids (5), benzene and substituted derivatives (4), carnitine (4), organoheterocyclic compounds (4), vitamins (4), and others (21). Additionally, 59, 56, and 47 metabolites were exclusively found in quail, chicken, and duck eggs, respectively."

       RE:  We have removed this paragraph in 4.2 section to the 4.1 section and marked them as blue.

Reviewer 3 Report

The manuscript investigate the metabolite profiles of different types of eggs using quasi-targeted metbolomic analysis. The topic is interesting, and has novelty for publication in this journal. However, there are some issues and concerns to address. I would recommend the manuscript for publication after changes and clarification. The specific comments are listed below

Majors

- In section 1 (introduction), Please explain the importance of quasi-targeted metabolomics approach for difference of three poultry eggs. 

- This study seems to only use the novegene database for the identification of metabolites. The major metabolites should compare with their authentic compounds to make sure reliability. Therefore, the authors should explain how to make the reliable data by using the novogene database.

- Results and Discussion section should be rewrote and re-organized. Even though the results showed a good distinguishment between different groups, the specific differential metabolites need to be mentioned. For example, in Fig S2, nucleotide and its derivatives are not specific name of compounds but carnitine is a specific compound. In addition. What are the specific names of phospholipid and fatty acids? TCA cycle can be differential metabolites?

Minors

- In 2.1. section, please add the specific year and month for egg harvest.

- In 2.2. section, why did authors dilute to final concentration of extracts as 53% methanol?

- In 2.2. section, 15000 g -> 15,000 g

- In 2.5 section, R’s cor.mtset()? Please correct this to the specific name.

- Please explain the method of qualification in 2.4. section. Using internal standard? Even so, please add internal standard information in 2.3. section.  

The major limitation in this manuscript is that the clarity of presentation and the written English need to be improved. There are many typo. Please revise the corrected one.

Author Response

Majors

- In section 1 (introduction), Please explain the importance of quasi-targeted metabolomics approach for difference of three poultry eggs.

 RE:  Quasi-targeted metabolomics reveals the metabolic changes of three types of poultry eggs by detecting small molecule metabolites, such as amino acids, sugars, fatty acids, nucleotides, etc. Compared with genomics and transcriptome, Metabonomics analysis focuses on the actual metabolites in eggs, which can directly affect the quality of eggs.

- This study seems to only use the novegene database for the identification of metabolites. The major metabolites should compare with their authentic compounds to make sure reliability. Therefore, the authors should explain how to make the reliable data by using the novogene database.

 RE:  The novegene database including the Q1, Q3, RT (retention time), DP (declustering potential), and CE (collision energy) qualitative information. The original data contains the ion equivalency qualitative information of the database. Having SCIEX QTRAP 6500+and corresponding reagents enables the detection of targeted metabolites.

- Results and Discussion section should be rewrote and re-organized. Even though the results showed a good distinguishment between different groups, the specific differential metabolites need to be mentioned. For example, in Fig S2, nucleotide and its derivatives are not specific name of compounds but carnitine is a specific compound. In addition. What are the specific names of phospholipid and fatty acids? TCA cycle can be differential metabolites?

 RE:  Because the large number of metabolites in different groups and the large number of groups, therefore, the specific metabolites are not mentioned in the main text. In Fig S2, the categories of differential expression metabolites among different groups were showed, instead of specific metabolites. Here the carnitines indicated the class of carnitine, including isovalerylcarnitine, arachidonoylcarnitine, carnitine-C3, carnitine-C18, etc. In addition, phospholipid and fatty acids also indicated the classes of phospholipid and fatty acids. TCA cycle indicated the compounds involved in TCA cycle. We have corrected them in the main text and in the figures.

Minors

- In 2.1. section, please add the specific year and month for egg harvest.

 RE:  We have added the year and month for egg harvest.

- In 2.2. section, why did authors dilute to final concentration of extracts as 53% methanol?

 RE:  In order to reduce matrix effect, better detect metabolites, and optimize peak shape, the samples were diluted to a methanol content of 53%.

- In 2.2. section, 15000 g -> 15,000 g

  RE:  We have revised the 15,000 g to 15000 g in the main text.

- In 2.5 section, R’s cor.mtset()? Please correct this to the specific name.

  RE:  We have revised “…cor.mtest()…” to “…cor.mtest …” in 2.5.

- Please explain the method of qualification in 2.4. section. Using internal standard? Even so, please add internal standard information in 2.3. section. 

RE:  Quasi-Targeted metabolomics uses multiple reaction monitoring mode (MRM) to detect experimental samples. The principle is that after ionization, the substance enters the triple quadrupole mass spectrometry system, and the Q1 quadrupole will screen the mother ions with a specific mass charge ratio (m/z), The screened parent ions enter the Q2 fragmentation pool and undergo a certain collision energy (CE) to break into sub ions with different mass to charge ratios. These sub ions then enter the Q3 quadrupole for further screening to obtain sub ions with specific mass to charge ratios for subsequent detection.

Comments on the Quality of English Language

The major limitation in this manuscript is that the clarity of presentation and the written English need to be improved. There are many typo. Please revise the corrected one.

      RE:  We have correct the typo in the main text.

Reviewer 4 Report

The authors did an experiment on the metabolite differences of three poultry eggs. Overall, I found the article interesting; however, there are some consideration

My comments are presented below.

1. Please, indicate the sufficient information about selected eggs (eg. species). In the manuscript, it says “There were six eggs for per egg type”. I can not find more information about chicken, quail, and duck eggs.

2. Add the information of “R package Pheatmap” in the section of 2.5 Data analysis.

3. It must be revised “Using R's cor.mtest (), statistically~~” in 2.5 Data analysis. Please check the parentheses.  

It is fine throughout the manuscript.

Author Response

  1. Please, indicate the sufficient information about selected eggs (eg. species). In the manuscript, it says “There were six eggs for per egg type”. I can not find more information about chicken, quail, and duck eggs.

       RE:  We have revised “There were six eggs for per egg type” to “Six chicken eggs, six duck eggs and six quail eggs were harvested respectively. These eggs of different poultry were harvested in October 2022.”

  1. Add the information of “R package Pheatmap” in the section of 2.5 Data analysis.

        RE:  We have added the information of “R package Pheatmap” in the section of 2.5.

  1. It must be revised “Using R's cor.mtest (), statistically~~” in 2.5 Data analysis. Please check the parentheses.  

       RE:  We have revised “…cor.mtest()…” to “…cor.mtest …” in 2.5.

Comments and Suggestions for Authors

In abstract, give the full names of QY, YW, HY, DY, DW, Q, D and H, in which the full names have never been indicated in the current manuscript and should be provided in the section of Materials and Methods. Furthermore, main findings should be given in abstract.

       RE:  We have added the full form for the abbreviations and listed them in glossary section. The main findings have given in abstract.

Reviewer 5 Report

In abstract, give the full names of QY, YW, HY, DY, DW, Q, D and H, in which the full names have never been indicated in the current manuscript and should be provided in the section of Materials and Methods. Furthermore, main findings should be given in abstract.

The authors should note that there are many issues to be improved for publication, but this reviewer do not want to point out all of them or surprise the current research.

The below issues are critical for the scientific publication of the current study;

The pairwise comparisons provided in Table 1 is not necessary because overall metabolic comparisons would be more useful for exploring metabolic differences among the eggs. These expressions of the results made them hard or limited to understand through Figures 3-5. Also, the authors mainly provided the differential metabolites of various eggs and their relations with metabolism, but information on quantitative results of the main metabolites in both egg albumin and egg yolk among the various eggs should be provided. Indeed, the discussion on the results is poor and thus should be in detail.

no comments.

Author Response

The pairwise comparisons provided in Table 1 is not necessary because overall metabolic comparisons would be more useful for exploring metabolic differences among the eggs. These expressions of the results made them hard or limited to understand through Figures 3-5. Also, the authors mainly provided the differential metabolites of various eggs and their relations with metabolism, but information on quantitative results of the main metabolites in both egg albumin and egg yolk among the various eggs should be provided. Indeed, the discussion on the results is poor and thus should be in detail.

RE:  We think that the Table 1 provides an overall display of the distribution of different metabolites among different groups. It can be retained.

       We have added the metabolites with the most significant differences for HY vs HW, QY vs QW, Hy vs HW, Q vs H, H vs D and Q vs D .
